# Small-Large Collaboration: Training-efficient Concept Personalization for Large VLM using a Meta Personalized Small VLM

## Abstract

Personalizing Vision-Language Models (VLMs) to transform them into daily assistants has emerged as a trending research direction. However, leading companies like OpenAI continue to increase model size and develop complex designs such as the chain of thought (CoT). While large VLMs are proficient in complex multimodal understanding, their high training costs and limited access via paid APIs restrict direct personalization. Conversely, small VLMs are easily personalized and freely available, but they lack sufficient reasoning capabilities. Inspired by this, we propose a novel collaborative framework named Small-Large Collaboration (SLC) for large VLM personalization, where the small VLM is responsible for generating personalized information, while the large model integrates this personalized information to deliver accurate responses. To effectively incorporate personalized information, we develop a test-time reflection strategy, preventing the potential hallucination of the small VLM. Since SLC only needs to train a meta personalized small VLM for the large VLMs, the overall process is training-efficient. To the best of our knowledge, this is the first training-efficient framework that supports both open-source and closed-source large VLMs, enabling broader real-world personalized applications. We conduct thorough experiments across various benchmarks and large VLMs to demonstrate the effectiveness of the proposed SLC framework. The code will be released.

## 1 Introduction

Recently, personalizing Vision-Language Models (Wang et al., 2024b; Liu et al., 2023; Hurst et al., 2024) (VLMs) that assist users' daily life has gained increasing interest. For example, VLMs should be aware of user-provided concepts such as a pet and generate personalized output including concept identifiers such as $\langle Bob \rangle$ or $\langle Lina \rangle$. To seamlessly integrate concepts into VLMs, many techniques make different attempts, including finetuning-based (Alaluf et al., 2024; Nguyen et al., 2024; 2025; An et al., 2024; 2025), Retrieval-Augmented Generation (RAG)-based (Hao et al., 2025), and representation-based (Pi et al., 2024) methods. While effective, deploying them at scale for all users presents critical challenges due to huge training costs (see Figure 1), particularly for large models. Taking finetuning-based methods as an example, existing approaches primarily focus on fine-tuning open-source VLM for personalization. However, it brings high training costs for fine-tuning large

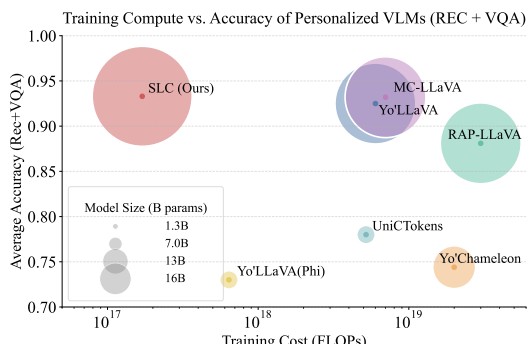

Figure 1: **Cost–size–accuracy of personalized VLMs.** Bubbles plot total training cost ($x$) vs. mean Rec.(Recognition)+VQA accuracy on the Yo'LLaVA datset ($y$); bubble area scales with model size. Our SLC (MetaC-3B + LLaVA-13B version) achieves the best accuracy (0.933) with $\sim 10^2\times$ lower training cost than 13B models.

models. Furthermore, leading companies like OpenAI limit access through paid APIs, also challenging existing paradigms. In contrast, small models can be easily personalized and are freely available, but they have limited reasoning ability.

Thus, a new paradigm is necessary, raising an important question: can we synthesize the benefits of both large and small models while mitigating their respective limitations?

While the small–large model collaboration paradigm is under-explored in the field of VLM personalization, several works (Chinchali et al., 2021; Zhang et al., 2022; Chen et al., 2025a; Li et al., 2025) have implemented it in other fields. However, most (Wang et al., 2024a; Ding et al., 2024) focus on reducing inference rather than training cost, which is a core bottleneck for VLM personalization. Wu et al. (2025) also proposed an insightful paradigm that small models solely handle the generation of chain-of-thought and large models only take responsibility for reasoning. However, in the context of VLM personalization, the model often produces hallucinations, and simply combining the outputs of the small and large models may not be effective.

Considering the above-mentioned challenges, and to support personalization for closed-source models, we propose a novel paradigm called Small-Large Collaboration (SLC) for personalizing large VLMs using small VLMs, illustrated in Figure 2.

Our key insight involves a clear division of tasks: small models manage user-specific perception, while large models facilitate reflection and general reasoning. To reduce training costs, we strategically train a meta personalized small VLM that can adaptively output various personalized information without the need for tuning. We pre-trained a small set of LoRA adapters for the small model; during inference, the most relevant adapters are activated dynamically, enabling zero-shot personalization for new user concepts. By leveraging the advantages of this paradigm, we can effectively combine small and large models, using small models to support the personalization of both open-source and closed-source large models. Additionally, our modular design naturally supports privacy-preserving hybrid architectures, in which lightweight local models perform personalized detection and interact securely with powerful cloud-hosted reasoning models. However, limited by model capability, small models are prone to output hallucinations, which may confuse concepts. Thus, we propose a reflection mechanism that the large model re-queries each detected concept with two focused yes/no VQA checks and suppresses any evidence it cannot visually confirm. Combined together, these two components allow SLC to fully leverage the reasoning power of large VLMs while also gaining the deployment advantages of small VLMs.

We summarize our contributions as follows:

- We propose a novel and efficient paradigm to personalize large VLMs using small VLMs, supporting both open-source and closed-source large VLMs personalization.

- We train a meta personalized small VLM and design a test-time reflection mechanism to reduce training costs and minimize potential hallucinations.

- Extensive experiments show that SLC achieves competitive performance across VLM personalization benchmarks, paving the way for real-world applications.

## 2 RELATED WORK

**Vision-Language Models**   Despite Vision Language Models (VLMs) (Bai et al., 2025; Hurst et al., 2024; Comanici et al., 2025) have demonstrated remarkable capabilities, the widespread adoption of VLMs for personalized applications is hampered by challenges. The sheer size and computational requirements of these models make it difficult and expensive to customize them for each user. To address the challenges of efficient deployment, researchers in the VLM field have explored several avenues. For instance, Qwen2.5-VL-3B (Bai et al., 2025) achieves lightweight through a highly optimized model architecture and a multi-stage training strategy. Meanwhile, explorations with models like Phi (Microsoft et al., 2025) series using meticulously curated data build highly capable models with far fewer parameters. However, these approaches are designed to create an efficient general-purpose model and are not transferable to our scenario, which requires personalized customization for every individual. In this work, we propose the SLC framework that pairs a small, efficient VLM for user-specific concepts learning with a large VLM for high-quality generation. This collabora-

tive paradigm resolves the inherent trade-off between personalization ability and the efficiency of training and deployment.

**Personalization of VLMs** Personalization in VLMs involves adapting models to recognize and incorporate user-specific concepts into their outputs for personalized interactions. MyVLM (Alaluf et al., 2024) augments pre-trained VLMs with training external concept heads, while Yo'LLaVA (Nguyen et al., 2024) uses learnable soft prompts to embed concepts by fine-tuning. To overcome the limitations of a single concept, MC-LLaVA (An et al., 2024) employs joint instruction tuning to handle multiple concepts, while RAP (Hao et al., 2025) adopts retrieval-augmented generation to enrich personalized responses. However, these methods require extensive training (as shown in Figure 1), creating a trade-off between personalization abilities and training costs. Facing this challenge, we propose a small-large collaboration paradigm to balance response quality and personalization efficiency.

**Small–Large Model Collaboration** There has been growing interest in collaborations between large and small models, primarily as a strategy to balance high performance with efficiency in general-purpose scenarios. Prevailing paradigms often focus on generic performance enhancement. For instance, the pipeline approaches (Lv et al., 2025; Zhang et al., 2024) use a small model to generate initial candidates for a large model to refine. More dynamic hybrid or routing strategies (Wang et al., 2025; Ong et al., 2025; Varangot-Reille et al., 2025; Chen et al., 2025b) employ a router to intelligently delegate tasks based on complexity. Similarly, auxiliary/enhancement paradigms (Shao et al., 2025) have been developed where one model assists another to improve overall performance. A particularly inspiring approach is the Cache of Thought (CoT) framework (Wu et al., 2025) that leverages a large model's knowledge to efficiently empower a small model. Whereas existing approaches have focused only on enhancing performance in general-purpose scenarios, our work is inspired by the CoT framework to address the challenge of personalization. In our paradigm, a lightweight small model acts as a "personalizer", detecting user-specific concepts and providing information to a large model to enable the generation of higher-quality, personalized responses.

Figure 2: **Inference pipeline of SLC**: a. Test-time detection by the small VLM $\mathcal{M}_s$; b. Test-time reflection by the large VLM $\mathcal{M}_l$; c. Answer generation by the large VLM $\mathcal{M}_l$.

## 3 METHOD

The method section is organized into two parts: 1) We first formalize the task of personalizing VLMs and state the system's objectives. We then outline our Small–Large Collaboration (SLC) inference pipeline, showing how the small VLM $\mathcal{M}_s$ and the large VLM $\mathcal{M}_l$ collaborate at test time. 2) We present the details of a meta-personalized small VLM and discuss the test-time reflection of a large VLM.

## 3.1 PROBLEM SETUP

The user registers a concept set $\{C_i^{\mathrm{u}}\}_{i=1}^N$—for example, $\langle$Bo$\rangle$ and $\langle$Lina$\rangle$. Each $C_i^{\mathrm{u}}$ with reference images $\mathcal{I}_{C_i^{\mathrm{u}}}$ and a brief textual description $\mathcal{T}_{C_i^{\mathrm{u}}}$ (e.g., "$\langle$ Bo $\rangle$ is a golden-retriever dog; it is my first pet dog."). At every dialogue turn, given an image–question pair $(I_t, q_t)$, the system must **(i)** identify which $C_i^{\mathrm{u}}$ appear in $I_t$, and **(ii)** answer any question about them.

## 3.2 SLC INFERENCE PIPELINE

### 3.2.1 OVERVIEW

Figure 2 sketches SLC's three-stage pipeline. In summary, our SLC framework addresses personalized VLM reasoning for large VLMs by utilizing small VLMs. A small model ($\mathcal{M}_s$) rapidly embeds user-defined concepts and produces structured, concept-level cues, while a powerful large model ($\mathcal{M}_l$) verifies those cues at test time and generates the final answer. Concretely, the collaboration consists of three sequential stages:

**a. Test-time detection.** At the beginning of each dialogue turn, the lightweight, meta-trained small VLM $\mathcal{M}_s$ embeds all registered user concepts and then examines $I_t$, providing concept-level cues $R_t = \{r_{t,i}\}_{i=1}^N$, where $r_{t,i} = \{\text{present}, \text{loc}_{\mathrm{abs}}, \text{loc}_{\mathrm{rel}}\}$.

**b. Test-time reflection.** For every $C_i^{\mathrm{u}}$ detected by $\mathcal{M}_s$, the large VLM $\mathcal{M}_l$ performs self-VQA checks to verify claims and sanitize cues, yielding $\tilde{R}_t$.

**c. Answer generation.** Finally, $\mathcal{M}_l$ takes $(I_t, q_t, \tilde{R}_t)$ and generates the final response $a_t$ to the user.

### 3.2.2 TEST-TIME REFLECTION OF LARGE VLM

**Concept-Level Cue Generation** To prime the test-time reflection step, $\mathcal{M}_s$ first emits a set of structured concept-level cues for the current step $t$:

$$R_t = \{r_{t,i}\}_{i=1}^N, \qquad r_{t,i} = \{\text{present}, \text{loc}_{\mathrm{abs}}, \text{loc}_{\mathrm{rel}}\}.$$

Here $N$ is the number of user-registered concepts, and each triple $r_{t,i}$ contains present — a Boolean flag indicating whether concept $C_i^{\mathrm{u}}$ is detected in image $I_t$; $\text{loc}_{\mathrm{abs}}$ — a string describing the concept's absolute position in the image (e.g., "upper-left corner", "center"); $\text{loc}_{\mathrm{rel}}$ — a string giving the concept's position relative to salient objects (e.g., "to the right of the cat", "behind the car"). When run in isolation, the small VLM $\mathcal{M}_s$ may hallucinate, confusing similar textures or backgrounds with a registered concept (see Figure 2). During test-time reflection, $\mathcal{M}_l$ cross-checks each $r_{t,i}$ against the image and the concept description $\mathcal{T}_{C_i^{\mathrm{u}}}$, producing a refined cue set $\tilde{R}_t$ with fewer hallucinations.

**Identity Extraction and Verification** For each personalized concept $C_i^{\mathrm{u}}$ detected by $\mathcal{M}_s$ in $R_t$, the large model $\mathcal{M}_l$ first extracts an immutable identity phrase $\mathrm{ID}(C_i^{\mathrm{u}})$ from its textual description $\mathcal{T}_{C_i^{\mathrm{u}}}$ (e.g., "a rabbit-like anime character"). It then performs two binary verifications: whether $\mathrm{ID}(C_i^{\mathrm{u}})$ appears at the absolute location $\text{loc}_{\mathrm{abs}}$ in the image, and whether it is consistent with the relative relation $\text{loc}_{\mathrm{rel}}$. The resulting answers $(a_1, a_2) \in \{\text{yes}, \text{no}\}^2$ are used to refine the cue $r_{t,i}$, where a double "no" sets present $= 0$, a single negative discards the corresponding absolute or relative location, and affirmative responses retain $r_{t,i}$ unchanged. This yields the refined cue set $\tilde{R}_t = \{\tilde{r}_{t,i}\}$, which is subsequently fused with $(I_t, q_t)$ for final answer generation by $\mathcal{M}_l$.

## 3.3 META-PERSONALIZED SMALL VLM

Current fine-tuning practices for different concepts lead to linear increases in training time and storage, which restrict the large-scale deployment of personalized applications. To tackle this issue, we propose a meta-training strategy for personalizing VLMs, inspired by the adaptation of T2I models (Rombach et al., 2022; Topal et al., 2025; Ruiz et al., 2024). The small VLM $\mathcal{M}_s$ is trained once offline; at test time, it incorporates concepts by simply selecting pre-learned adapters—no optimization necessary, which further minimizes overall training expenses.

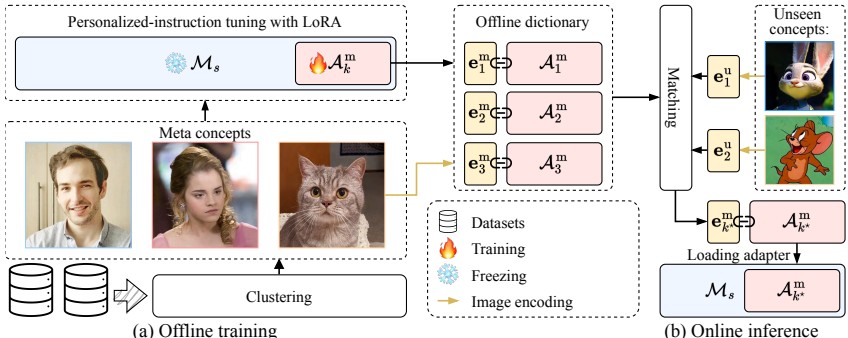

Figure 3: **Overview of our meta-personalization framework for the small VLM.** (a) During offline training, we cluster concept images into $K$ meta concepts and apply LoRA tuning to train $K$ corresponding adapters $\{\mathcal{A}_k^{\mathrm{m}}\}_{k=1}^{K}$. Each meta concept is encoded as $(\mathbf{e}_k^{\mathrm{m}}, \mathcal{A}_k^{\mathrm{m}})$ and is stored offline. (b) At test time, a new concept is encoded into $\mathbf{e}_j^{\mathrm{u}}$ and matched to its closest meta concept. The matched adapter is loaded into $\mathcal{M}_s$ for the downstream task, enabling tuning-free inference.

### 3.3.1 OFFLINE TRAINING

As shown in Figure 3(a), we extract CLIP embeddings for images in several public personalization datasets and run K-means clustering to form $K$ appearance clusters. The centroid of each cluster defines a meta-concept $C_k^{\mathrm{m}}$, yielding the set $\{C_k^{\mathrm{m}}\}_{k=1}^{K}$. These meta-concepts span a broad semantic spectrum—covering humans, animals, and diverse objects—and can be adapted or combined to represent new, semantically related ideas within the same category. For each centroid, we train a Low-rank adaptation (LoRA) (Hu et al., 2022) adapter $\mathcal{A}_k^{\mathrm{m}}$ using the personalized-instruction tuning recipe from prior work (An et al., 2024) (training details in the Appendix). We store every meta-concept embedding $\mathbf{e}_k^{\mathrm{m}}$ together with its adapter, forming an offline dictionary.

### 3.3.2 ONLINE INFERENCE

At run time, the user may register multiple new concepts $\{C_i^{\mathrm{u}}\}_{i=1}^{N}$. As Figure 3(b) illustrates, we first compute an embedding for each concept by averaging its reference-image features, then form a scenario embedding $\bar{\mathbf{e}}^{\mathrm{u}} = \frac{1}{N} \sum_{i=1}^{N} \mathbf{e}_i^{\mathrm{u}}$. We select a single meta-adapter for the scenario via the cosine rule

$$k^{\star} = \arg\max_{k} \cos\left(\bar{\mathbf{e}}^{\mathrm{u}}, \mathbf{e}_k^{\mathrm{m}}\right). \tag{1}$$

The chosen adapter $\mathcal{A}_{k^{\star}}^{\mathrm{m}}$ is plugged into $\mathcal{M}_s$ and used to detect all registered concepts. Since no weights are updated online, the pipeline remains tuning-free while scaling to an unlimited number of new concepts.

## 4 EXPERIMENT

### 4.1 EXPERIMENTAL SETUP

**Evaluation Datasets**  All experiments are conducted on the MC-LLaVA (An et al., 2024) and Yo'LLaVA (Nguyen et al., 2024) datasets. MC-LLaVA is a multi-concept personalization dataset for VLMs comprising 118 diverse concepts that are grouped into 40 scenarios by data source. The concepts span real-world personalities, anime characters, real objects, and anime-style objects. MC-LLaVA provides both single- and multi-concept test sets for recognition, VQA, and text-only QA. Yo'LLaVA contains 40 distinct concepts covering objects, buildings, and people, and supplies single-concept test sets for the same three tasks.

We observed that several personalization methods tend to overfit to a concept's reference images, paying less attention to the current visual input and occasionally hallucinating. To probe this issue, we additionally construct a Special Question–Answer (SQA) set: targeted VQA queries on Yo'LLaVA test images whose correct answers contradict spurious cues present in the concept's training data, forcing the model to ground its reasoning on the image at hand. Representative SQA examples are provided in the Appendix.

**Implementation Details**   To build the meta-concept dictionary, we merge all concepts in the Yo'LLaVA and MC-LLaVA training splits. These concepts are clustered into $K = 10$ meta-concepts (see Appendix for details). To avoid data leakage, any concepts (along with images and Q&As) associated with the meta-concept are removed from every evaluation set. We adopt Qwen2.5-VL-3B-Instruct (Bai et al., 2025) as the backbone of $\mathcal{M}_s$ and meta-train it to obtain our Meta-Concepts-Model-3B (MetaC-3B). Each meta-concept adapter is optimized with LoRA for 80 steps using a learning rate of $5 \times 10^{-5}$ and a batch size of 64 on 8 A800 GPUs. To reduce randomness, we run the experiment three times and report the average.

Table 1: **Performance comparison of personalized VLMs on Yo'LLaVA and MC-LLaVA datasets.** For SLC, $\mathcal{M}_s$ = MetaC-3B. The **best** and second-best performances are highlighted.

| Method | Training cost | Yo'LLaVA dataset | | | | MC-LLaVA dataset | | | | |
| | | Rec. | VQA | Text-only | SQA | Rec. | | | VQA | Text-only |
| | FLOPs | Weight | Acc | Acc | Acc | Single | Multi | Weight | Acc | Acc |
|---|---|---|---|---|---|---|---|---|---|---|
| Image prompt + GPT-4o | - | 0.901 | 0.915 | 0.891 | 0.850 | 0.831 | 0.823 | 0.827 | 0.904 | 0.733 |
| Text prompt + GPT-4o | - | 0.872 | 0.930 | 0.871 | 0.900 | 0.746 | 0.822 | 0.781 | 0.889 | 0.702 |
| SLC ($\mathcal{M}_l$=GPT-4o) | $1.7 \times 10^{17}$ | **0.951** | **0.979** | **0.895** | **0.900** | 0.760 | **0.931** | 0.830 | **0.937** | **0.739** |
| SLC ($\mathcal{M}_l$=LLaVA-1.5-13B) | $1.7 \times 10^{17}$ | 0.895 | 0.971 | 0.879 | 0.883 | 0.762 | 0.878 | 0.801 | 0.861 | 0.692 |
| MC-LLaVA | $7.0 \times 10^{18}$ | 0.947 | 0.934 | 0.885 | 0.725 | **0.912** | 0.845 | **0.878** | 0.890 | 0.709 |
| Yo'LLaVA | $6.0 \times 10^{18}$ | 0.924 | 0.929 | 0.883 | 0.713 | 0.744 | 0.729 | 0.737 | 0.655 | 0.658 |
| RAP-LLaVA | $3.0 \times 10^{19}$ | 0.845 | 0.917 | 0.874 | 0.813 | 0.747 | 0.688 | 0.713 | 0.784 | 0.685 |
| LLaVA + text prompt | - | 0.819 | 0.913 | 0.803 | 0.725 | 0.594 | 0.549 | 0.573 | 0.817 | 0.553 |

## 4.2 PERFORMANCE COMPARISON

### 4.2.1 COMPARED METHODS

We benchmark SLC against several representative personalization approaches for VLM on both the Yo'LLaVA and MC-LLaVA test sets:

- **Yo'LLaVA** (Nguyen et al., 2024): one of the earliest VLM personalization methods (built on LLaVA-1.5-13B). Because it natively supports only single-concept scenarios, we follow the multi-concept adaptation Yo'LLaVA-M in MC-LLaVA (An et al., 2024).

- **MC-LLaVA** (An et al., 2024): a VLM personalization method specifically designed for multi-concept personalization, also based on LLaVA-1.5-13B.

- **RAP-LLaVA** (Hao et al., 2025): a retrieval-augmented generation (RAG) approach for multimodal personalization, again built on LLaVA-1.5-13B.

- **Upper bound (GPT-4o)** (OpenAI et al., 2024): Personalized image or text prompts with test questions are fed to GPT-4o, serving as an optimistic performance ceiling thanks to its strong multimodal reasoning capacity.

- **Lower bound (LLaVA + text prompt)** (Liu et al., 2023): Each test question is paired with its corresponding personalized text prompt and evaluated on the vanilla LLaVA-1.5-13B, providing a conservative baseline.

For a fair comparison, we instantiate SLC with two large VLMs, LLaVA-1.5-13B and GPT-4o. During text-only QA, no image is supplied. Instead, the question is concatenated with the corresponding concept descriptions, and $\mathcal{M}_l$ directly generates the answer. Further implementation details for the text-only QA protocol are deferred to the Appendix.

### 4.2.2 RESULT ANALYSIS

Table 1 reports the overall comparison. Our evaluation metrics follow the protocols of the Yo'LLaVA (Nguyen et al., 2024) and MC-LLaVA (An et al., 2024) datasets. In particular, for the recognition task we adopt a weighted score defined as Weighted $= 0.5 \times$ Yes recall $+ 0.5 \times$ No recall, where *Yes recall* denotes the proportion of correctly predicted "yes" responses among existence queries with the concept present, and *No recall* analogously measures the proportion of correctly predicted "no" responses when the concept is absent. For the Yo'LLaVA dataset, we additionally

list the training FLOPs consumed by each method, so that accuracy can be assessed jointly with computational cost. Below, we highlight the superiority of SLC from three perspectives.

**(a) Superior accuracy without task-specific finetuning.** Across the two SLC variants, nearly every first– or second–place in Table 1 is occupied. SLC with $\mathcal{M}_l$=GPT-4o attains the top scores on all Yo'LLaVA metrics (0.951 Rec., 0.979 VQA, 0.895 Text-only) and secures either the highest or runner-up performance on MC-LLaVA, surpassing every other GPT-4o-based baseline. When $\mathcal{M}_l$ is replaced by LLaVA-1.5-13B, SLC still matches—or exceeds—prior finetuned methods, verifying that our meta-trained small model plus test-time reflection can unleash the strong reasoning power of large VLMs without additional finetuning.

**(b) Training efficiency.** SLC performs only one meta-training on $\mathcal{M}_s$, consuming $1.7 \times 10^{17}$ FLOPs—about $40\times$ less than the fine-tune–heavy Yo'LLaVA ($6.0 \times 10^{18}$) and MC-LLaVA ($7.0 \times 10^{18}$), and almost $200\times$ less than the retrieval-augmented RAP-LLaVA ($3.0 \times 10^{19}$). For Yo'LLaVA and MC-LLaVA, the training cost grows nearly linearly with the number of personalized concepts, whereas SLC's training cost is fixed after a single meta-training stage. Although RAP-LLaVA's expense is a one-off rather than linear, it remains orders of magnitude higher than SLC. These results confirm that SLC is more training-efficient.

**(c) Reduced over-fitting and hallucination.** On the SQA set—designed to expose memorization—SLC ties GPT-4o for the top score (0.900) and exceeds all finetuned methods by $>10$ pp. The gain attests to the synergy between our two-model collaboration and the test-time reflection step: $\mathcal{M}_s$ supplies structured concept cues, while $\mathcal{M}_l$ subsequently verifies those cues, suppressing spurious matches and grounding the answer in the current image.

These results indicate that a lightweight meta-personalized VLM $\mathcal{M}_s$ plus a powerful but frozen large VLM $\mathcal{M}_l$ is both more accurate and far cheaper to train than existing finetune-heavy pipelines.

### 4.3 ABLATION STUDY AND ANALYSIS

#### 4.3.1 EFFECT OF LoRA TRAINING AND PROMPTING STRATEGIES

To test how different prompt configurations during LoRA training and inference affect generalization to unseen concepts, we train adapters for $\mathcal{M}_s$ = Qwen2.5-VL-3B (Bai et al., 2025) on the Yo'LLaVA dataset.

During training, we prepend each question prompt in the QA pair with the concept description or leave it unchanged. At inference, we mirror this choice, yielding six settings in total, plus two baselines without LoRA. Results are reported in Table 2.

It is obvious that using concept descriptions consistently at both training and inference

Table 2: **Prompt strategies for LoRA training and inference on the Yo'LLaVA dataset.** TP. = text prompt. The **best** performances are highlighted.

| Training Setting | Inference Setting | Rec. | VQA | Text-only |
|---|---|---|---|---|
| | | Weight | Acc | Acc |
| w/o LoRA | w/ TP. | 0.714 | 0.813 | 0.810 |
| | w/o TP. | 0.508 | 0.786 | 0.655 |
| w/ TP. | w/ TP. | **0.819** | **0.835** | **0.843** |
| | w/o TP. | 0.603 | 0.791 | 0.697 |
| w/o TP. | w/ TP. | 0.744 | 0.818 | 0.733 |
| | w/o TP. | 0.562 | 0.790 | 0.702 |

stages yields the best overall performance on recognition, VQA, and text-only QA (0.819 / 0.835 / 0.843). When the description is included only during training, accuracy drops significantly at test time, likely due to a mismatch between training and inference prompts. In contrast, models trained without descriptions can still benefit from their inclusion at inference, though they fail to match the jointly prompted setting. Notably, all LoRA-trained variants outperform the non-adapted baselines, highlighting the effectiveness of lightweight adaptation. These results collectively indicate that prompt consistency is crucial for generalization to unseen concepts.

#### 4.3.2 EFFECT OF META-CONCEPT POOL SIZE AND TOP-K ADAPTER SELECTION

To investigate how the meta-concept pool size and Top-$K$ adapter fusion influence the performance of SLC, we evaluate two configurations on the Yo'LLaVA dataset recognition task: **(i)** the small model alone ($\mathcal{M}_s$ = MetaC-3B) and **(ii)** the complete SLC pipeline ($\mathcal{M}_s + \mathcal{M}_l$). For each setting, we vary **(a)** the number of meta-concepts obtained via $k$-means clustering of CLIP embeddings, and **(b)** the number of nearest meta-concept adapters Top-$K$ averaged at test time. Each meta-concept

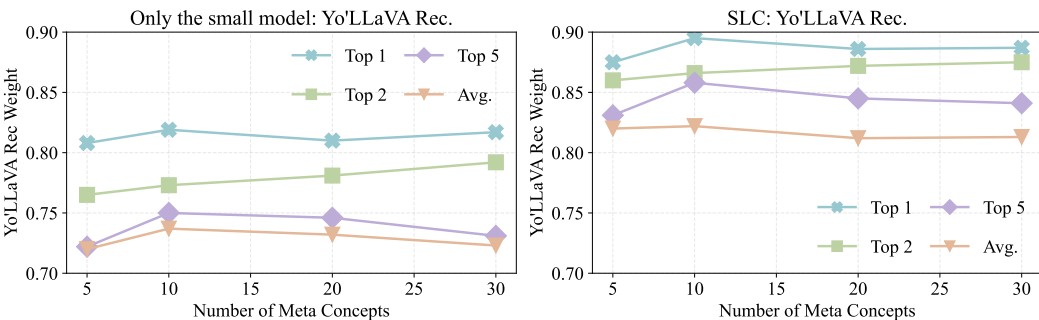

Figure 4: **Impact of meta-concept pool size and Top-$K$ adapter fusion on Yo'LLaVA recognition.** **Left**: the small model only ($\mathcal{M}_s$). **Right**: full SLC pipeline ($\mathcal{M}_s + \mathcal{M}_l$). Across all $K \in \{1, 2, 5, \text{all}\}$, the best accuracy occurs at 10 meta-concepts, and Top-1 selection consistently yields the highest scores.

has its own LoRA adapter; at inference, we retrieve the Top-$K$ closest meta-concepts to the queried concept and merge their adapters before decoding.

Figure 4 reveals two consistent trends. **First**, recognition peaks at 10 meta-concepts for both $\mathcal{M}_s$ and SLC, indicating an optimal balance between coverage and redundancy. **Second**, selecting a single adapter (Top-1) consistently surpasses Top-2 and Top-5, showing that fusing multiple adapters dilutes concept-specific cues. These findings validate our design principle: a moderately sized (10) adapter library combined with dynamic Top-1 selection yields the strongest performance.

Table 3: **Varying the small model $\mathcal{M}_s$ while fixing $\mathcal{M}_l$ = LLaVA-1.5-13B.** "Yes/No recall" measures the $\mathcal{M}_s$'s ability to correctly predict the presence/absence of a concept, respectively. Adding $\mathcal{M}_l$ improves all metrics and markedly boosts "No recall", evidencing hallucination reduction.

| $\mathcal{M}_s$ | VLM | Training data | Only $\mathcal{M}_s$ | | | | $\mathcal{M}_s + \mathcal{M}_l$ | | | |
| | | | Rec. | | | VQA | Rec. | | | VQA |
| | | | Yes recall | No recall | Weight | Acc | Yes recall | No recall | Weight | Acc |
| MetaC-3B | Qwen2.5VL-3B | 47.3k | 0.900 | 0.759 | 0.829 | 0.835 | 0.898 | 0.893 | 0.895 | 0.971 |
| RAP-Phi3-V | Phi3-V-3.8B | 260k | 0.912 | 0.754 | 0.833 | 0.866 | 0.906 | 0.896 | 0.901 | 0.971 |
| RAP-LLaVA | LLaVA-1.5-13B | 260k | 0.926 | 0.764 | 0.845 | 0.917 | 0.921 | 0.920 | 0.921 | 0.975 |
| Yo'LLaVA | LLaVA-1.5-13B | 190k | 0.949 | 0.898 | 0.924 | 0.929 | 0.947 | 0.928 | 0.936 | 0.971 |
| LLaVA + text prompt | LLaVA-1.5-13B | - | 0.734 | 0.903 | 0.819 | 0.913 | 0.734 | 0.929 | 0.832 | 0.966 |

### 4.3.3 VARYING THE SMALL MODEL

We further investigate how the capacities of $\mathcal{M}_s$ influence system behavior on the Yo'LLaVA dataset (Nguyen et al., 2024). To obtain more diverse results, we select five personalized models with different training and inference paradigms as the small model $\mathcal{M}_s$, each emphasizing distinct aspects: **MetaC-3B**: a meta-personalized model trained and inferred with an efficient meta-learning strategy; **RAP-Phi3-V** and **RAP-LLaVA** (Hao et al., 2025): two models sharing the same training data and RAG-based inference, but differing in the sizes of their VLM backbones; **Yo'LLaVA** (Nguyen et al., 2024): a conventional personalized model that trains and evaluates separately for each concept; **LLaVA+text prompt**: a training-free baseline built on LLaVA-1.5 (Liu et al., 2024), as detailed in Section 4.2.1. The experimental results are presented in Table 3. Across different choices of $\mathcal{M}_s$, we observe varying tendencies in Recognition: trained models generally achieve higher "Yes recall", while the training-free "LLaVA+text prompt" baseline tends to favor "No" predictions. For the first four trained models, false positives are relatively common. When comparing the performance of $\mathcal{M}_s$ alone versus $\mathcal{M}_s + \mathcal{M}_l$, two key findings emerge. **First**, $\mathcal{M}_l$ substantially reduces false positives, as evidenced by the consistent improvement in "No recall". **Second**, the overall performance of SLC is positively correlated with the intrinsic performance of $\mathcal{M}_s$ (as reflected in both Recognition and VQA), while being less sensitive to the specific training or inference paradigm adopted.

### 4.3.4 SCALING THE LARGE MODEL

We explore how the capacities of $\mathcal{M}_l$ affect SLC behavior on the Yo'LLaVA dataset (Nguyen et al., 2024). Table 4 reports the impact of varying the size of $\mathcal{M}_l$ (ranging from no $\mathcal{M}_l$ to 3B and up to 72B) on the performance of SLC. We observe that incorporating $\mathcal{M}_l$ substantially improves the performance of the small model on both Recognition and VQA tasks. Moreover, as the capacity of $\mathcal{M}_l$ increases, the overall performance of SLC continues to improve. This clearly demonstrates that, although $\mathcal{M}_s$ has limited capability, $\mathcal{M}_l$ can effectively correct the errors made by $\mathcal{M}_s$, highlighting a scaling law that characterizes the steady gains of our SLC framework with respect to $\mathcal{M}_l$.

### 4.3.5 ABLATING THE SLC PIPELINE

To pinpoint the roles of $R_t$ produced by $\mathcal{M}_s$ and the test-time reflection, we compare SLC against three ablated variants: w/o reflection, where $\mathcal{M}_s$ is enabled but $\mathcal{M}_l$ skips verification; w/o detector, where $\mathcal{M}_l$ performs reflection without any cues from $\mathcal{M}_s$; and Pure VLM, where both components are disabled, reducing SLC to "LLaVA + text prompt" in Table 1. Table 5 yields three findings:

Table 4: **Scaling $\mathcal{M}_l$ while fixing $\mathcal{M}_s$ = MetaC-3B.**

| $\mathcal{M}_l$ | Model Size | Rec. Weight | VQA Acc | Text-only Acc |
|---|---|---|---|---|
| - | - | 0.829 | 0.835 | 0.843 |
| Qwen2.5VL | 3B | 0.872 | 0.956 | 0.844 |
| Qwen2.5VL | 7B | 0.903 | 0.973 | 0.879 |
| Qwen2.5VL | 32B | 0.932 | 0.975 | 0.881 |
| Qwen2.5VL | 72B | 0.944 | 0.979 | 0.890 |

**(1) Synergy is essential.** The full SLC tops nearly every metric. Removing either element reduces performance (Yo'LLaVA VQA drops to 0.958 w/o reflection and 0.962 w/o $\mathcal{M}_s$), confirming that the two modules are essential. **(2) Reflection suppresses hallucination.** The "No recall" rate—correctly predicting a concept's absence—jumps from 0.814 (w/o reflection) to 0.893 with reflection, and reaches 0.905 when only reflection is present. Thus, $\mathcal{M}_l$'s verification is crucial for filtering false positives. **(3) $\mathcal{M}_s$ supplies precise cues.** Omitting $\mathcal{M}_s$ hurts "Yes recall" ($0.898 \rightarrow 0.767$) and pushes MC-LLaVA recognition weight down from 0.801 to 0.710, showing that the efficiency of $\mathcal{M}_s$.

Table 5: **Ablation of SLC components.** We fix $\mathcal{M}_s$ = MetaC-3B, $\mathcal{M}_l$ = LLaVA-1.5-13B, and selectively disable $\mathcal{M}_s$ or the test-time reflection. $\checkmark$ = enabled, $-$ = disabled.

| $\mathcal{M}_s$ | Test-time reflection | Yo'LLaVA dataset | | | | | MC-LLaVA dataset | | | |
|---|---|---|---|---|---|---|---|---|---|---|
| | | Rec. | | | VQA | SQA | Rec. | | | VQA |
| | | Yes recall | No recall | Weight | Acc | Acc | Single | Multi | Weight | Acc |
| $\checkmark$ | $\checkmark$ | 0.898 | 0.893 | 0.895 | 0.971 | 0.883 | 0.762 | 0.878 | 0.801 | 0.861 |
| $\checkmark$ | - | 0.841 | 0.814 | 0.827 | 0.958 | 0.838 | 0.748 | 0.642 | 0.705 | 0.843 |
| - | $\checkmark$ | 0.767 | 0.905 | 0.836 | 0.962 | 0.825 | 0.712 | 0.707 | 0.710 | 0.839 |
| - | - | 0.734 | 0.903 | 0.819 | 0.913 | 0.725 | 0.594 | 0.549 | 0.573 | 0.817 |

## 5 CONCLUSION

We propose SLC, a novel Small–Large Collaboration paradigm for personalizing VLMs, effectively balancing training efficiency and personalization capability. SLC combines a meta-trained, tuning-free small model for personalized cues generation with a large model performing test-time reflection to reduce hallucinations. This approach addresses the longstanding cost–performance trade-off and naturally supports personalization of both open-source and closed-source large VLMs. Its modular design further enables privacy-preserving hybrid deployments. Extensive experiments validate SLC's scalability, efficiency, and reliability, highlighting its potential for real-world applications.

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

## A    THE USE OF LLMS

During the coding and debugging phases, we utilized LLMs for technical guidance. After collaboratively drafting the manuscript, we once again turned to LLMs to enhance and refine its language and overall style.

## B    PROMPT TEMPLATES

### B.1    TEST-TIME DETECTION PROMPT TEMPLATES

The prompt template used by the small VLM $\mathcal{M}_s$ to detect all concepts provided by the user is shown in Table 6.

---

*System prompt.*
You are a high-precision concept detector.
**Task**
You should inspect the image and output **one** JSON object that **covers every** concept in the **Concept List** provided by the user while conforming to the schema below.
For each concept ⟨concept_id⟩ in the list:
  • If *visible*, set:
    • "present": `true`
    • "location-absolute": "concise area, e.g. "top-left quadrant"
    • "location-relative": "spatial relation, e.g. "to the left of the person in black suit"
  • If *not visible*, set:
    • "present": `false` → omit all other keys
The final output should be a JSON object like: {
  "⟨concept_id_1⟩": {
    "present": ⟨boolean⟩,
    "location-absolute": ⟨string⟩ or "",
    "location-relative": ⟨string⟩ or "",
  },
  "⟨concept_id_2⟩": {
    "present": ⟨boolean⟩,
    "location-absolute": ⟨string⟩ or "",
    "location-relative": ⟨string⟩ or "",
  }
...
}

**Rules**
  • Output plain English text only—no Markdown, no code fences.
  • Keep every concept ID enclosed in angle brackets (⟨⟩).
  • If present = `false`, omit all other keys.
  • Boolean literals must be lowercase `true`/`false`.
  • Do not add any extra keys, comments, or explanatory text.

*User prompt.*
**Concept List**
$\{C_i^u : \mathcal{T}_{C_i^u}\}_{i=0}^N$

---

Table 6: **Test-time detection prompt** for $\mathcal{M}_s$.

### B.2    TEST-TIME REFLECTION PROMPT TEMPLATES

Test-time reflection is performed in two concise steps by the large VLM:

  • **Identity extraction** (Table 7): $\mathcal{M}_l$ extracts each concept's category and attributes.

- **Self-VQA verification** (Table 8): $\mathcal{M}_l$ uses the category extracted in the previous step to answer *yes/no* questions, thereby confirming the absolute and relative locations of each concept reported as visible by $\mathcal{M}_s$.

---

*System prompt.*
You are an information extractor.
**Task**
Inspect the textual descriptions below and return **one** JSON object that **covers every** concept in the **Concept List** while conforming to the schema:
- "category" : permanent class, e.g. "a golden retriever puppy", "a blue cartoon character"
- "attributes" : mutable traits, e.g. "always playful expression; dresses in trendy clothes"

**Example**
Example:
*User prompt*
   **Concept List**:
   ⟨bo⟩: ⟨bo⟩ is a cute golden retriever puppy with a playful expression.
   ⟨shiba-sleep⟩: ⟨shiba-sleep⟩ is a shiba inu sleeping peacefully in a cozy home.
*Expected output*
{
  "⟨bo⟩": {
    "category": "a golden retriever puppy",
    "attributes": "always playful expression"
  },
  "⟨shiba-sleep⟩": {
    "category": "a shiba inu",
    "attributes": "can sleep peacefully; lives in cozy home"
  }
}

**Rules**
- Output plain English text only—no Markdown, no code fences.
- Keep every concept ID enclosed in angle brackets (⟨⟩).
- Provide **exactly** the two keys for each concept—no extras.
- Do not add comments or explanatory text outside the JSON object.
*User prompt.*
**Concept List**
$\{C_i^u : \mathcal{T}_{C_i^u}\}_{i=0}^N$

---

Table 7: **Identity extraction prompt** for $\mathcal{M}_l$.

---

*System prompt.*
You are a visual verifier.
**Task**
You should answer each visual question with **yes** or **no**—nothing else.

*User prompt.*
Q1. Do you see $\{\text{ID}(C_i^u)\}$ at {location-absolute} of the image? (yes or no)
Q2. Is $\{\text{ID}(C_i^u)\}$ {location-relative}? (yes or no)

**Rules**
- Provide exactly one "yes" or "no" per question.
- If there are $N$ questions, output $N$ tokens separated by a single space.
- Do not include any additional words, punctuation, or commentary.

---

Table 8: **Test-time reflection prompt** for the large VLM $\mathcal{M}_l$ for each concept detected by the small VLM.

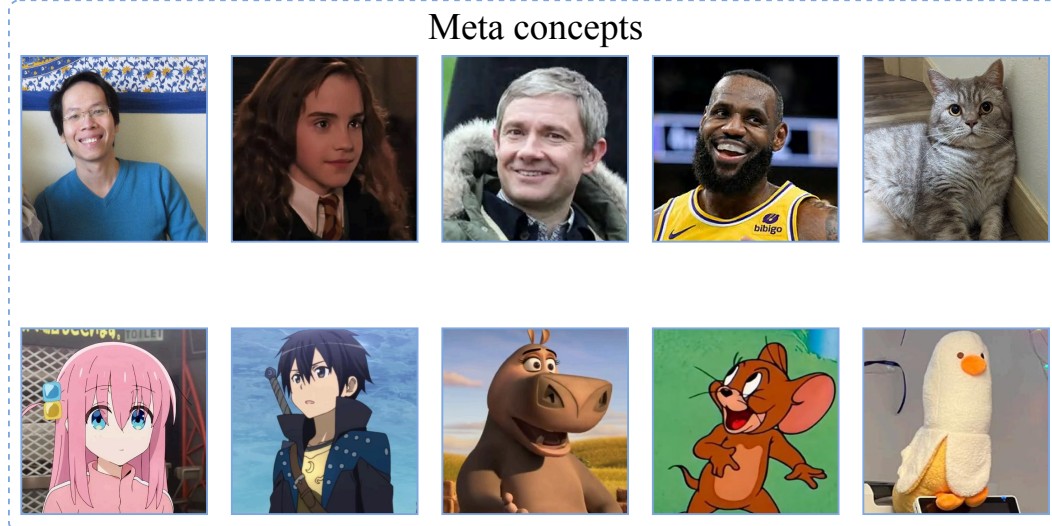

Figure 5: **Visualizations of the 10 meta-concepts.**

### B.3 ANSWER GENERATION PROMPT TEMPLATES

The large VLM $\mathcal{M}_l$ produces the final response by conditioning on the structured detection report returned from the previous stages. The concise prompt is shown in Table 9.

---

*System prompt.*
**Detection Report**
$\{C_i^u: \text{present, category, attributes, location-absolute, location-relative}\}_{i=1}^{N}$

**Rules**
Use the Detection Report to answer the user's visual question.
   • **category**: immutable essence (e.g. "a golden retriever puppy").
   • **attributes**: mutable traits that may or may not be visible (e.g. "always playful expression").
   • If present = `false`, it means the concept is not in the image. You should not mention the concept; reply "no" if asked about its presence.
   • If present = `true`, it means it concept is in the image. You should ground your answer strictly on the provided fields; reply "yes" if asked about its presence.

*User prompt.*
{User prompt}

---

Table 9: **Answer-generation prompt** for $\mathcal{M}_l$.

## C LORA TRAINING DETAILS

### C.1 META-CONCEPT CLUSTERING & VISUALIZATION

**CLIP Embedding Clustering Pipeline** To extract meta-concepts from the Yo'LLaVA (Nguyen et al., 2024) and MC-LLaVA (An et al., 2024) datasets, we employ a pipeline based on feature clustering. First, we generate a feature embedding for each image using a pre-trained CLIP (Radford et al., 2021) model. We then create the vector representation for each concept by computing the average of its corresponding image embeddings. These concept-level vectors are subsequently clustered using the K-Means algorithm with the number of clusters set to 10. This process yields 10

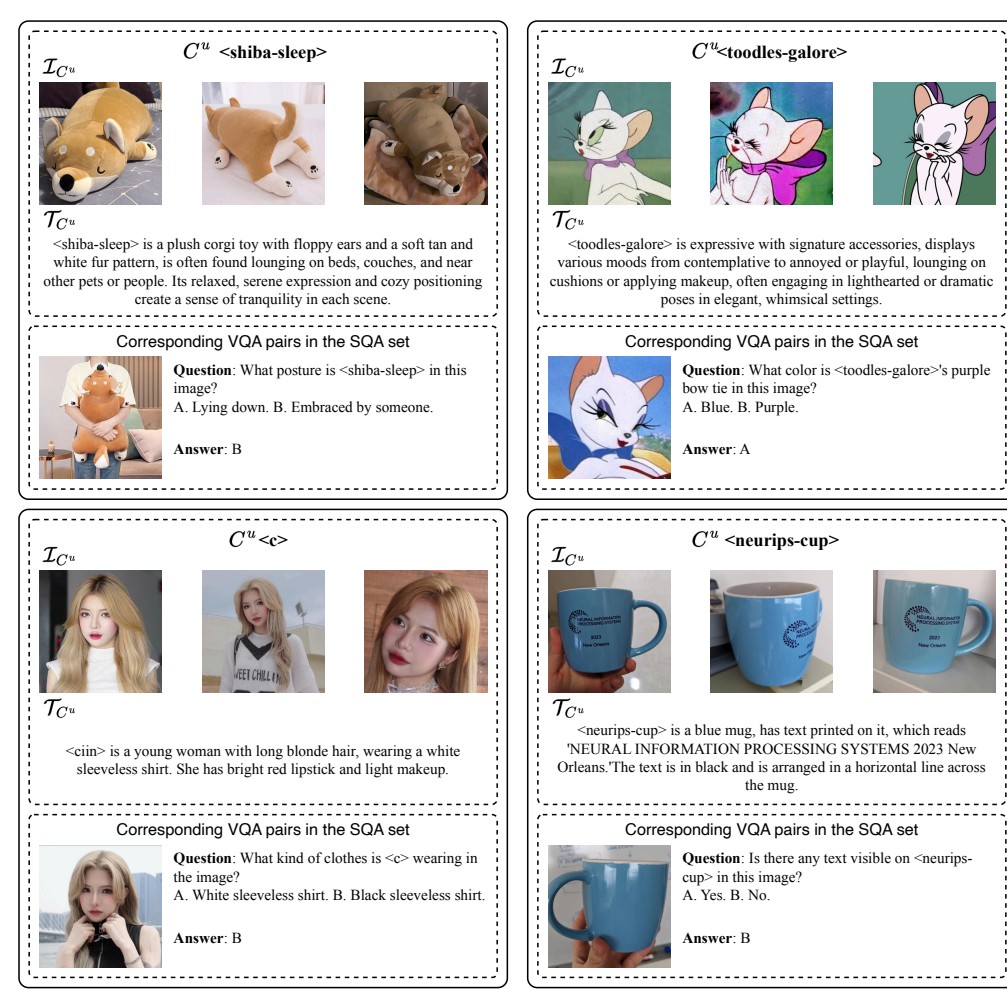

Figure 6: **Examples of the Special Question-Answer (SQA) set.**

meta-concepts, each defined as the concrete concept within a cluster whose embedding is closest to the cluster's centroid.

**Visualizations of the 10 Meta-Concepts** Figure 5 provides a visualization of the 10 meta-concepts derived from our pipeline, exhibiting diversity and clear separation. The visualization showcases a wide array of categories, ranging from humans and animals to objects and cartoon characters.

### C.2 TRAINING CONFIGURATION

We fine-tune the Qwen2.5-VL-3B-Instruct (Bai et al., 2025) model using Low-Rank Adaptation (LoRA) (Hu et al., 2022). LoRA modules of rank $r = 8$ and scaling factor $\alpha = 8$ are injected into every self-attention projection and FFN linear layer. Each meta-concept adapter is trained for 80 steps using the AdamW optimizer with a learning rate of $5 \times 10^{-5}$. The training is distributed across 8 A800 GPUs with a total batch size of 64 (8 samples per GPU).

## D  DATASET AND EVALUATION DETAILS

**Special Question–Answer (SQA) Set Examples** To diagnose overfitting, we build a Special Question–Answer (SQA) set composed entirely of VQA pairs. For every concept $C_u$ in the Yo'LLaVA test split, we create at least two questions whose correct answers contradict visual or

textual cues found in the concept's own training data ($\mathcal{I}_{C_u}$, $\mathcal{T}_{C_u}$). A model must therefore ground its reasoning on the test image itself—rather than memorised artefacts—to succeed. Representative examples are shown in Figure 6.

**Text-Only QA Protocol**    Although SLC is designed for personalized VQA, it also accommodates text-only questions. Following RAP-MLLM (Hao et al., 2025), we concatenate the user query with the concept description $\mathcal{T}_{C_u}$. Before concatenation, however, $\mathcal{M}_l$ performs identity extraction (Table 7) to convert each description into a structured category/attributes field. The resulting prompt — {structured category/attributes field + user question} — is then fed back to $\mathcal{M}_l$ for answering. Table 1 in the main text shows that this extra step yields a slight accuracy gain on the text-only test set.

# E    ADDITIONAL CASE STUDIES

We present three case studies. All examples use MetaC-3B as the small VLM ($\mathcal{M}_s$) and LLaVA-1.5-13B (Liu et al., 2023) as the large VLM ($\mathcal{M}_l$). Figure 7 showcases a straightforward success case, whereas Figure 8 and Figure 9 highlight how test-time reflection corrects a false positive from the small VLM.

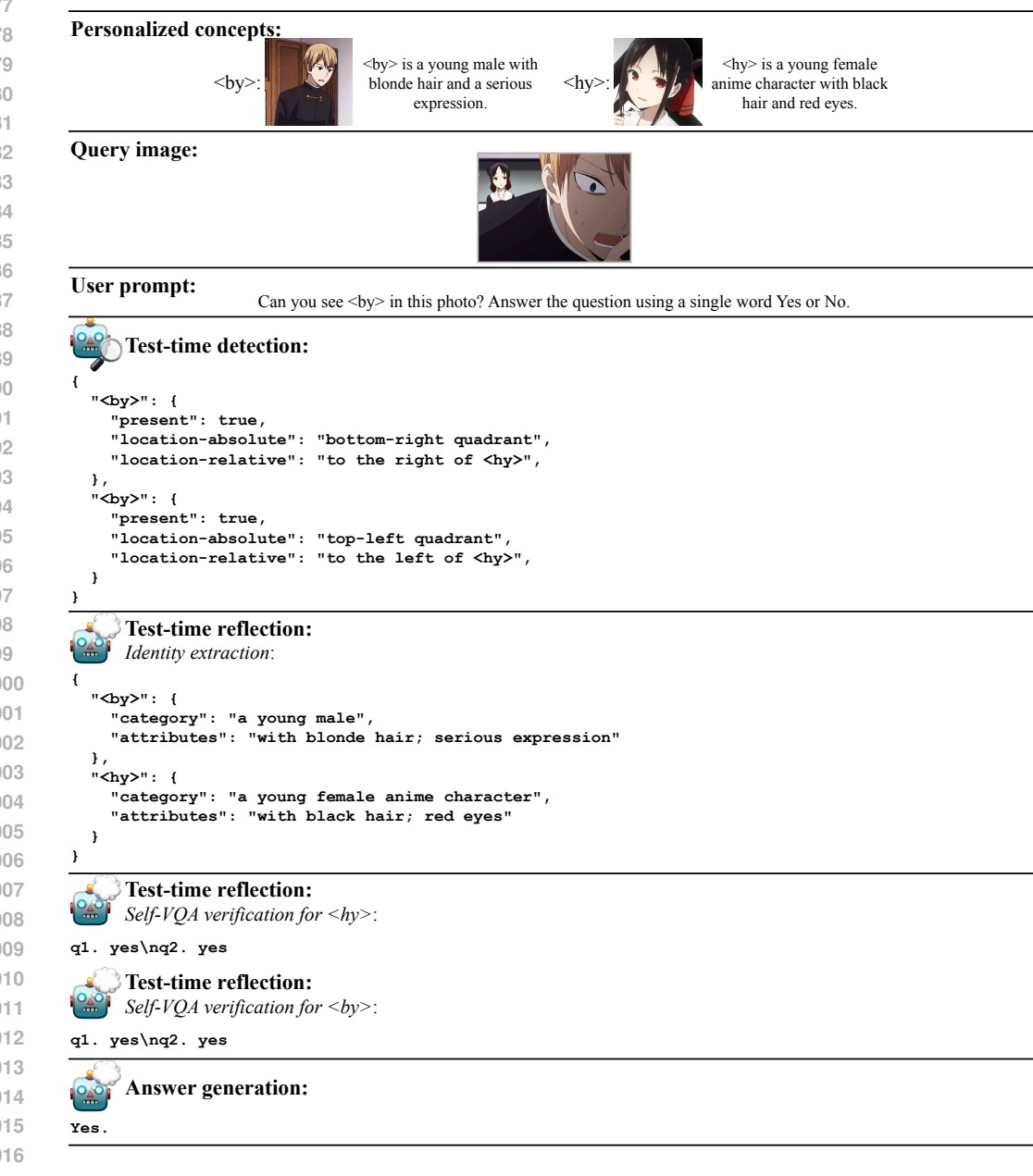

Figure 7: **End-to-end inference example of the SLC pipeline.** The small VLM correctly identifies all concepts, the large VLM finds no inconsistencies, and SLC answers the user's question accurately.

**Personalized concepts:**

<tr>: 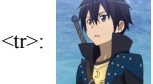 <tr> is a young male with black hair, wearing a black jacket with a white fur-lined collar and a white emblem on the left side of his chest.

<ys>: 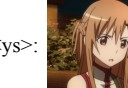 <ys> is a young female anime character with long orange hair.

**Query image:**

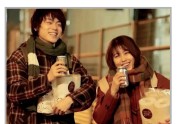

**User prompt:**

What is <ys> and <tr> doing in this image?

**Test-time detection:**

```
{
  "<tr>": {
    "present": false,
    "location-absolute": "",
    "location-relative": "",
  },
  "<ys>": {
    "present": true,
    "location-absolute": "bottom-right quadrant",
    "location-relative": "to the right of a male",
  }
}
```

**Test-time reflection:**
*Identity extraction*:

```
{
  "<tr>": {
    "category": "a young male",
    "attributes": "has black hair; wearing a black jacket with a white fur-lined collar and a
white emblem on the left side of his chest"
  },
  "<ys>": {
    "category": "a young female anime character",
    "attributes": "with long orange hair"
  }
}
```

**Test-time reflection:**
*Self-VQA verification for <ys>*:

```
q1. no\nq2. no
```

**Answer generation:**

```
<ys> and <tr> are not present in this image.
```

Figure 8: **End-to-end inference example of the SLC pipeline.** The small VLM mistakenly detects ⟨ys⟩, but the error is rectified by the large VLM via test-time reflection, enabling SLC to provide the correct answer.

**Personalized concepts:**

<butin>:

<butin> is a cream-colored Siberian Husky with striking blue eyes, relaxes comfortably in various indoor scenes, from couches to wooden floors. Known for its playful and laid-back demeanor, this Husky lounges in different positions, sometimes upside-down, showcasing its fluffy coat and cream-colored paws with an amusing, regal charm.

**Query image:**

**User prompt:**

Can you check if <butin> is in this photo?Answer the question using a single word Yes or No.

**Test-time detection:**

```
{
  "<butin>": {
    "present": true,
    "location-absolute": "top-left quadrant",
    "location-relative": "on top of the person's lap",
  }
}
```

**Test-time reflection:**

*Identity extraction*:

```
{
  "<butin>": {
    "category": "a cream-colored Siberian Husky with striking blue eyes",
    "attributes": "playful and laid-back demeanor; fluffy coat; cream-colored paws; regal charm"
  }
}
```

**Test-time reflection:**

*Self-VQA verification for <butin>*:

```
q1. no\nq2. no
```

**Answer generation:**

```
No.
```

Figure 9: **End-to-end inference example of the SLC pipeline.** The small VLM mistakenly detects ⟨butin⟩, but the error is rectified by the large VLM via test-time reflection, enabling SLC to provide the correct answer.

