# OpenReview forum: "Small-Large Collaboration: Training-efficient Concept Personalization for Large VLM using a Meta Personalized Small VLM"
_ICLR.cc/2026/Conference — Submitted to ICLR 2026_

### Official Review · Reviewer_mZas · 2025-10-24

**Soundness:** 2
**Presentation:** 3
**Contribution:** 2
**Rating:** 4
**Confidence:** 2

**Summary:**

This paper addressed the high training cost and difficulty in adapting closed-source models in the personalization process of large visual language models (VLMS), and proposed the Small-Large Collaboration (SLC) framework. The structured information (existence, absolute/relative position) of the user concept is generated through the meta-personalized small VLM, and then the small model illusion is filtered through the "reflection during testing" (double yes/no VQA verification) by the large VLM, and finally the personalized response is generated. To enhance training efficiency, a meta-personalized small VLM is designed by clustering the concepts of the public dataset through K-Means to obtain "meta-concepts", pre-training the corresponding LoRA adapter, and dynamically matching the user concepts with the meta-concepts during testing to achieve zero-tuning personalization.

**Strengths:**

1. The modular architecture supports "local small models processing personalized information + cloud large models inference", adapting to edge device scenarios and providing a feasible path for the implementation of VLM personalization.
2. The experiments cover multiple types of tasks and compare mainstream methods, and provide scalability experiments for small models and large models.

**Weaknesses:**

1.The core idea relies on the combination of existing technologies, with no breakthrough design, i.e. meta-personalization is based on K-Means clustering and LoRA (both are mature technologies).
2. The closed-source model only tested GPT-4o. Small models only verify 3B-level models such as Qwen2.5-VL-3B and Phi3-V, and do not test small models below 1B. The value of the meta-concept K is only fixed at 10. The performance changes of values such as K=5 and 15 have not been analyzed, lack of confirming whether the selection of K is robust.
3.The hyper-parameters of LoRA training (rank=8, learning rate = 5e-5, step count = 80) did not undergo sensitivity analysis, and the rationality of parameter selection could not be proved.
4. The two-step process of "small model checking + large model reflection" in SLC may increase the reasoning delay compared to directly fine-tuning the large model, but the paper did not provide a comparison of reasoning speeds.
5.The performance changes in multi-user and multi-concept scenarios (such as processing over 100 user concepts simultaneously) have not been explored, lack of proving the scalability of the framework. Moreover, "privacy protection" is only a conceptual design and has not been tested in experimental study.

**Questions:**

1.Why is K=10 determined to be Optimal? Could the performance-cost curves of K=5, 10, 15, and 20 be supplemented to prove the robustness of K=10? If the K value changes, will the accuracy of the meta-personalized small VLM fluctuate significantly?
2.Testing only one closed-source model, GPT-4o, can supplement the experimental results of others? If the API cannot be obtained, can the core conditions to adapt to different closed-source models (such as API input format and output constraints) be analyzed theoretically?
3.The paper emphasizes training efficiency but does not mention the time consumption of reasoning. Could you supplement the comparison in terms of "single-sample inference time" and "number of samples processed per second"? Especially in multi-concept scenarios, will the two-step process lead to a significant increase in reasoning latency?
4.Why choose "absolute position + relative position" dual-dimensional Verification? Could the ablation results of "absolute location only", "attribute matching only", and "location + attribute" be supplemented to demonstrate the optimality of the two-dimensional validation?
5. Can the performance changes of small model parameters from 0.5B to 3B be supplemented to verify the adaptability under different resource constraints?
6. When the modular design mentions "privacy protection", can it be supplemented by actual tests (such as the information leakage risk of local small models and the data transmission volume of local-cloud interaction) to prove the effectiveness of this design in real privacy scenarios?

---

### Official Review · Reviewer_qVDV · 2025-10-26

**Soundness:** 2
**Presentation:** 3
**Contribution:** 2
**Rating:** 4
**Confidence:** 4

**Summary:**

This paper presents a novel Small-Large Collaboration (SLC) framework for training-efficient personalization of large Vision-Language Models (VLMs). The core idea involves using a meta-personalized small VLM to generate concept-level cues, which are then verified and refined by a large VLM through a test-time reflection mechanism. The authors demonstrate that SLC achieves competitive performance on standard benchmarks (Yo'LLaVA and MC-LLaVA) while significantly reducing training costs compared to existing fine-tuning-based approaches. The framework supports both open-source and closed-source large VLMs and offers a modular design suitable for privacy-aware deployments.

**Strengths:**

1. Novel Collaboration Paradigm: The idea of leveraging a small VLM for personalized concept detection and a large VLM for reflection and reasoning is innovative and timely. It effectively addresses the trade-off between training cost and model capability.

2. Training Efficiency: The meta-personalized small VLM, trained only once with LoRA adapters, enables zero-shot adaptation to new user concepts without additional tuning. This results in orders-of-magnitude reduction in training FLOPs.

3. Comprehensive Evaluation: The paper includes thorough experiments across multiple benchmarks, model scales, and ablation studies, demonstrating the effectiveness and scalability of the proposed method.

4. Reflection Mechanism: The test-time reflection step helps mitigate hallucinations from the small VLM, improving the robustness and accuracy of the final responses.

**Weaknesses:**

1. **Inadequate Handling of False Negatives from the Small VLM:**
The current framework only applies test-time reflection to concepts where the small VLM outputs `present = true`. However, for concepts marked as `present = false`, the large VLM performs no further verification. This may lead to missed detections (false negatives) and lower recall, especially when the small VLM fails to recognize a concept due to limited generalization or semantic drift.


2. **Limited Validation of Meta-Concept Generalization:**
The meta-personalized small VLM relies on a fixed set of only \( K = 10 \) meta-concepts derived from Yo'LLaVA and MC-LLaVA datasets. While the authors explore the impact of pool size and Top-\(K\) selection, the evaluation is confined to these datasets. It remains unclear how well the method generalizes to unseen or highly diverse user concepts in real-world settings.


3. **Reliability Assumption of Large VLM Verification:**
The test-time reflection mechanism assumes that the large VLM (e.g., GPT-4o or LLaVA-13B) can accurately perform the VQA-based verification. However, large VLMs themselves are prone to hallucinations and errors, particularly in complex visual scenes. The paper does not evaluate the accuracy of this verification step, which could lead to error propagation—for example, the large VLM incorrectly confirming a false positive from the small VLM.


4. **Incomplete Computational Cost Analysis:**
 While the paper emphasizes training efficiency (low FLOPs), it overlooks the inference-time costs. The large VLM must perform two VQA checks for each concept detected by the small VLM, which can lead to significant latency when dealing with multiple concepts. There is no quantitative analysis of inference time, memory usage, or resource consumption.


5. **Clarity on the Role of the Small VLM:**
While the small VLM is tasked with generating concept cues, the heavy reliance on the large VLM for verification raises questions about the small model's indispensable contribution. A more detailed ablation or comparison with a large-VLM-only baseline would help clarify the unique value added by the small VLM.

**Questions:**

Please see the weaknesses above.

1. The authors should consider extending the reflection mechanism to also validate a subset of `present = false` concepts—for instance, based on confidence scores or semantic relevance—to improve recall. Additionally, evaluation metrics should explicitly report recall rates for both presence and absence of concepts.

2. The authors should validate the meta-concept approach on additional datasets or real-user scenarios to assess its robustness. Experiments with varying \( K \) and adapter fusion strategies in more diverse settings would strengthen the claims of generalizability.

3. The authors should assess the reliability of the large VLM's verification through human evaluation or ground-truth validation. Incorporating confidence scoring or majority voting across multiple verifications could help mitigate this issue.

4. The authors should provide measurements of inference latency and computational overhead under varying numbers of concepts.

---

### Official Review · Reviewer_v9zg · 2025-10-27

**Soundness:** 3
**Presentation:** 2
**Contribution:** 2
**Rating:** 4
**Confidence:** 4

**Summary:**

This paper proposes a framework named Small-Large Collaboration (SLC) for personalizing Vision-Language Models (VLMs). The framework combines a small, meta-trained VLM and a large VLM to achieve efficient personalization. The small VLM is used to generate personalized information, while the large VLM integrates this information to generate personalized responses. The authors also introduce a test-time reflection strategy to prevent hallucination from the small VLM. The framework is designed to be training efficient and supports both open-source and closed-source large VLMs.

**Strengths:**

1. It's reasonable to use small VLM to reduce the training cost, since not all tasks require the full capacity of a large VLM.
2. The proposed test-time reflection strategy effectively reduces hallucination from the small VLM, which help the large VLM to achieve better performance.
3. The experimental results demonstrate that the SLC framework can achieve competitive performance with significantly reduced training costs compared to traditional fine-tuning methods.
4. The framework can support closed-source large VLMs.

**Weaknesses:**

1. The paper could improve in organization. Most of the training details of the meta-personalized small VLM and the construction of the proposed SQA dataset are deferred to the appendix, making it difficult for readers to fully understand the implementation of the SLC framework. The authors should include more implementation details of both components in the main text to improve readability and reproducibility.
2. The proposed meta-personalization strategy is similar to existing works like "Meta-LoRA" (Topal et al., 2025). The paper should better clarify the technical differences between the proposed method and prior works, and ideally provide a fair empirical comparison to demonstrate its advantage.
3. The framework may lead to higher computational costs during inference since both the small and large VLMs are invoked multiple times. This could limit its practicality in real-time or resource-constrained scenarios. The authors should discuss the trade-off between training and inference efficiency.
4. The experiments are limited to recognition and question answering tasks. It remains unclear how the proposed framework performs on other multimodal tasks such as image captioning or personalized conversation, which are also important for real-world personalization.
5. Some implementation aspects remain unclear. For instance, what is the exact input format or prompt template for the small VLM?

**Questions:**

1. What's the input format for the small VLM? How does it achieve zero-shot personalization?
2. In Table 3, what's the 190k training data for Yo'LLaVA?

---

### Meta-Review · Area_Chair_4TMw · 2026-01-07

**Summary:**

The paper proposes Small-Large Collaboration, a framework designed for efficient personalization of Vision-Language Models (VLMs). The core mechanism involves a small, meta-trained VLM that generates personalized concept-level cues, which are then verified by a large VLM through a "test-time reflection" strategy to mitigate hallucinations. By using K-Means clustering to create "meta-concepts" and pre-training corresponding LoRA adapters, the framework achieves zero-shot personalization for new user concepts.

Reviewers recognize the efficiency gains in training and the practical value of a modular architecture that supports both open-source and closed-source models. However, the reviewers collectively rate the paper as "marginally below the acceptance threshold" due to concerns regarding inference latency, the robustness of the proposed strategy, and the lack of implementation details in the main text.

**Reviewer Concerns:**

No Rebuttal.

**Reviewer Scores:**

No Rebuttal.

---

### Decision · Program_Chairs · 2026-01-26

Reject